# Advancing aquaculture: Production of xenogenic catfish by transplanting blue catfish (*Ictalurus furcatus*) and channel catfish (*I. punctatus*) stem cells into white catfish (*Ameiurus catus*) triploid fry

**Darshika Udari Hettiarachchi**[1]*, **Veronica N. Alston**[1], **Logan Bern**[1], **Jacob Al-Armanazi**[1], **Baofeng Su**[1], **Mei Shang**[1], **Jinhai Wang**[1], **De Xing**[1], **Shangjia Li**[1], **Matthew K. Litvak**[2], **Rex A. Dunham**[1], **Ian A. E. Butts**[1]

1 School of Fisheries, Aquaculture and Aquatic Sciences, Auburn University, Auburn, Alabama, United States of America, 2 Department of Biology, Mount Allison University, Sackville, New Brunswick, Canada

\* darshikahettiarachchi@gmail.com

**Data Availability Statement:** All data are in the manuscript and/or supporting information files.

## Abstract

Xenogenesis has been recognized as a prospective method for producing channel catfish, *Ictalurus punctatus* ♀ × blue catfish, *I. furcatus* ♂ hybrids. The xenogenesis procedure can be achieved by transplanting undifferentiated stem cells derived from a donor fish into a sterile recipient. Xenogenesis for hybrid catfish embryo production has been accomplished using triploid channel catfish as a surrogate. However, having a surrogate species with a shorter maturation period, like white catfish (*Ameiurus catus*), would result in reduced feed costs, labor costs, and smaller body size requirements, making it a more suitable species for commercial applications where space is limited, and as a model species. Hence, the present study was conducted to assess the effectiveness of triploid white catfish as a surrogate species to transplant blue catfish stem cells (BSCs) and channel catfish stem cells (CSCs). Triploid white catfish fry were injected with either BSCs or CSCs labeled with PKH 26 fluorescence dye from 0 to 12 days post hatch (DPH). No significant differences in weight and length of fry were detected among BSCs and CSCs injection times (0 to 12 DPH) when fry were sampled at 45 and 90 DPH (P > 0.05). The highest survival was reported when fry were injected between 4.0 to 5.5 DPH (≥ 81.2%). At 45 and 90 DPH, cell and cluster area increased for recipients injected from 0 to 5.2 DPH, and the highest cluster area values were reported between 4.0 to 5.2 DPH. Thereafter, fluorescent cell and cluster area in the host declined with no further decrease after 10 DPH. At 45 DPH, the highest percentage of xenogens were detected when fry were injected with BSCs between 4.0 to 5.0 and CSCs between 3.0 to 5.0 DPH. At 90 DPH, the highest number of xenogens were detected from 4.0 to 6.0 DPH when injected with either BSCs or CSCs. The current study demonstrated the suitability of white catfish as a surrogate species when BSCs and CSCs were transplanted into triploid white catfish between 4.0 to 6.0 DPH (27.4 ± 0.4°C). Overall, these findings allow enhanced efficiency of commercializing xenogenic catfish carrying gametes of either blue catfish or channel catfish.

**Funding:** Agriculture and Food Research Initiative Competitive Grant no. 2018-67015-27614 from the USDA National Institute of Food and Agriculture.

**Competing interests:** The authors have declared that no competing interests exist. The authors declare that they have no known competing financial interests or personal relationships that could have appeared to influence the work reported in this paper.

## Introduction

Catfish farming stands as the dominant sector in the United States aquaculture industry, topping out at 157 million pounds produced annually and accounting for ~62% of all United States aquaculture production in 2019 [1]. Until recently, channel catfish, *Ictalurus punctatus*, was the primary species cultured for food fish [2]. However, it has shifted toward the production of hybrid catfish, produced by crossing the channel catfish, ♀ with the blue catfish, *I. furcatus* ♂ due to improved performance in feed conversion, disease resistance, dissolved oxygen tolerance, fillet yield, dress-out percentage, growth, and harvestability compared with both parent species [3–5]. Although the hybrid catfish industry has seen sustained growth, there are still challenges due to the lack of natural hybridization between these two species. Artificial fertilization is the only viable technology for commercial hybrid catfish embryo production, which still has disadvantages, such as sacrificing mature blue catfish males for sperm collection, hormonal stimulation of females, extra holding space, special handling techniques, additional feed, and labor cost [6, 7]. Hence, new innovative reproductive technologies such as xenogenesis can be used to mitigate some of the challenges in hybrid catfish production.

Xenogenesis is a method of reproduction in which successive generations differ from each other and no genetic material is transmitted from the surrogate to the offspring [8]. This current innovative method addresses major challenges in hybrid catfish production by offering an alternative approach for artificial fertilization by facilitating natural or semi-natural spawning between a normal channel catfish female and a xenogenic channel catfish male. Xenogenesis can be achieved by transplanting undifferentiated germ cells, such as primordial germ cells (PGCs), spermatogonial stem cells (SSCs), or oogonial stem cells (OSCs) originated from a donor fish into a sterile recipient fish which acts as a surrogate [9–13]. PGCs, SSCs, and OSCs possess self-renewal ability, making them useful as donor cells for xenogenesis transplantation. After transplantation, these cells can migrate, colonize and proliferate, giving rise to female or male germ cells, after which the gonad differentiates into an ovary or a testis, depending on the hosts' genetic sex [14]. Due to these unique characteristics, significant work has been devoted to transplant blue catfish SSCs and OSCs for xenogenic applications to support hybrid catfish production [12, 15–17].

However, the xenogenic process is not yet optimized. In all these experiments triploid channel catfish was used as the surrogate species during stem cell transplantation. Channel catfish usually reach sexual maturity in 2 to 4 years [18]. It is advantageous to use a surrogate species that has a short maturation time which can reduce the feed cost, labor cost, holding space, and ease of handling. Hence, utilization of white catfish (*Ameiurus catus*) as a surrogate is potentially advantageous as it has a shorter maturation time and small size, improving the potential of xenogenesis for commercial-scale hybrid catfish production.

According to Dunham et al. [19] and Fobes [20], white catfish mature in 1 to 2 years compared to 2 to 4 years for channel catfish. White catfish begin to reach sexual maturity as early as 15.2 cm in fork length in northern California populations [21] and up to 23.0 cm total length in New Jersey populations [22]. Spawning of white catfish could be observed in early May when the temperature of the water ranges from 19 to 21˚C [21]. In addition, white catfish grow rapidly as fingerlings, but their growth slows when they approach sexual maturation at 1 year of age [19]. Furthermore, Fobes [19] reported that using luteinizing hormone-releasing hormone analog (LHRHa) implants at 90 μg/kg in domestic white catfish resulted in 100% ovulation of the females after 72 hours. Thus, white catfish have several desirable characteristics compared to channel catfish as a surrogate species for the xenogenic process. Due to these major advantages, the present study aimed to identify the suitability of triploid white catfish as a surrogate species and to identify the best age to transplant blue and channel catfish stem cells to produce xenogenic white catfish.

## Materials and methods

### Ethics statement

All investigations and experimental studies on animals were conducted according to the Institutional Animal Care and Use Committee (IACUC) protocols and guidelines. The protocols were approved by IACUC chairperson Dr. Terry Brandebourg (PRN Number: 2021–3893). Furthermore, research team members completed Auburn University-certified training sessions on Biosafety, Fish Handling, Animal Emergency Care Plans, and Biosecurity for animal areas before commencing the current study.

### Broodstock management and triploid induction

Broodstock was cultured in 0.04 ha earthen ponds averaging 1 m in depth at the EW Shell Fisheries Center at Auburn University, AL, USA. Fish were fed to satiation with 32% protein pellet feed five days per week during summer and three days per week during winter. In addition, the fish were fed 36% protein broodstock feed two months prior to spawning. During the broodstock collection period, DO, temperature, pH, TAN, nitrite, nitrate, hardness, and alkalinity levels were 6.9 ± 1.7 mg/L, 27.0 ± 2.1°C, 7.1 ± 0.3, <0.5 mg/L, <0.5 mg/L, <20 mg/L, 89.5 ± 5 mg/ L, and 35.5 ± 10 mg/L, respectively. Gravid white catfish females (n = 3; mean ± SEM weight = 0.7 ± 0.2 kg) and sexually mature males (n = 3; mean ± SEM weight = 0.6 ± 0.3 kg) were collected by seining the pond with a 3.8 cm mesh seine net.

Gravid females were administered intraperitoneal injections of luteinizing hormone releasing hormone analogue (LHRHa) at 90 μg/kg body weight. Then, white catfish females and males (1: 1 ratio) were placed in 60 L aquaria in a calm environment. After 48 h, each tank was carefully observed. When the females began to ovulate (few eggs could be seen at the bottom of the tank) the males and females were separated. Males were immediately euthanized by blunt force trauma to the center of the head followed by pithing as found in section S6.2.2 of the 2020 American Veterinary Medical Association (AVMA) guidelines for euthanasia to ensure no effects of tricaine methanesulfonate (MS-222) on sperm viability. Then, the sperm collection was conducted according to the standard protocol described by Dunham and Masser [23]. In brief, testes were surgically removed with a sterile scalpel and forceps, rinsed with 0.9% saline solution to remove blood, and then gently dried. Testes were minced with a scalpel blade and macerated to generate a sperm/testes solution which was then filtered with a 100 μm mesh. Ten mL of 0.9% saline was added for each 1 g of testes to dilute the sperm sample.

The ovulated females were anesthetized with 100 ppm tricaine MS-222 and buffered with 100 ppm sodium bicarbonate. Eggs were manually stripped into spawning pans that had been lined with Crisco commercial vegetable oil and fertilized with a prepared sperm solution [15, 16]. A Fullers' earth (MP Biomedicals, Santa Ana, CA) solution was added to the sperm-egg solution to start the fertilization process and to prevent eggs from sticking together. Fuller's earth solution was prepared by adding 6 g of powdered Fuller's earth to 1 L of pond water. After 3 min, triploid induction was carried out according to standard protocols [12, 15–17]. Briefly, the fertilized eggs were transferred into a steel chamber and after an additional 2 min hydrostatic pressurize was applied at 7,000 PSI for 5 min. After pressure shock, the eggs were removed from the chamber and placed in a trough supplemented with $CaCl_2$ at 50 ppm for water hardening before incubating. After 1 h, eggs were incubated in hanging mesh baskets (7.0 m × 0.4 m × 0.2 m) placed in flow-through hatching troughs which were aerated with paddle wheels. Ploidy level was verified by measuring erythrocyte nuclei volume with a Coulter-counter at the USFWS Warm Springs Regional Fisheries Center, Warm Springs, GA [24].

## Extraction of donor stem cells from juvenile blue and channel catfish

On a daily basis, four sexually immature channel catfish (mean ± SEM length = 25.3 ± 5.9 cm, weight = 399.5 ± 100.3 g) and four blue catfish (mean ± SEM length = 33.5 ± 5.2 cm, weight = 488.2 ± 84.3 g) were selected (n = 4) and immediately euthanized by blunt force trauma to the center of the head followed by pithing, to avoid any potential effects on cell viability caused by MS-222. After being euthanized and gonad extraction, blue catfish stem cell (BSCs) and channel catfish stem cell (CSCs) isolation were performed according to the standard protocols described by Hettiarachchi et al., Shang et al., and Abualreesh et al. [15, 25–29]. In brief, blue and channel catfish gonads were separately placed on a sterile petri dish (100 mm × 15 mm) containing 5 mL of Hanks' Balanced Salt Solution [(HBSS, SH30048.24, GE Healthcare Life Sciences) supplemented with 1.0 μg/mL NaHCO$_3$ (Church & Dwight Co., NG) and 100 unit/mL penicillin—streptomycin (115140–122, Life Technologies)]. Samples were cleaned with a freshly prepared 0.5% bleach solution for 2 min, then further cleaned with HBSS and phosphate-buffered saline (PBS; J62701, Alfa Aesar). These samples were minced with a sterilized scalpel blade and the trypsinization process was conducted by adding 0.25% trypsin—ethylenediamine tetraacetic acid (EDTA; 25243–261, Life Technologies) at 50 times the weight of each sample. After the trypsinization process, each cell suspension was filtered using 70 μm and 40 μm cell strainers (352364 /382398, VWR International) and centrifuged at 500 g (Eppendorf Centrifuge 5418 R) for 10 min. Then, the resulting pellet was resuspended in 2 mL of Dulbecco's Modified Eagle's Medium/DMEM [DMEM (10–173-CV, VWR International)] supplemented with 10% fetal bovine serum (FBS; 10438044, Life Technologies), 100 unit/mL penicillin-streptomycin (15140–164, Life Technologies), and 200 mM L-glutamine (A2916993, Life Technologies).

## Labeling the stem cells with PKH26 and transplantation

Extracted stem cells were marked with PKH26 red fluorescence cell linker (CGLDIL, Sigma-Aldrich, St. Louis, MO) following the guidelines provided by the manufacturer. CSCs and BSCs were transplanted into triploid channel catfish fry daily from 0 to 12 days post-hatch (DPH) with stem cells (channel/ blue) labeled with either PKH26 or without PKH26, according to standard methods described by Hettiarachchi et al. [15, 16]. In brief, each day white catfish triploid fry (three replicates for each treatment / 30 fry per treatment) were anesthetized by placing them in a petri dish with 10 mg/L tricaine methanesulfonate (MS-222) buffered with 10 mg/L NaHCO$_3$ solution. Fry were observed microscopically at 1.5× (Amscope, Irvine, CA) and injected with 1 uL of unsorted cell suspension containing 80,000 cells through a 33-gauge needle (outer diameter: 0.209 mm; inner diameter: 0.108 mm; Hamilton, Reno, NV) with a manual microinjector (Hamilton, Reno, NV). Fry were positioned on their side and the needle gently inserted in the cavity between the anal fin and yolk sac where the genital ridge was expected to be formed (Fig 1). After injection, the fry were placed in a recovery container. During the cell transplantation period, the average water temperature was 27.4 ± 0.4˚C.

## Growth and survival

Stem cell transplanted fry were transferred back to fry cages (0.20 m × 0.20 m × 0.25 m) in a recirculating system at a density of 30 fry per cage. Initially, fry were fed a standard, powdered commercial catfish fry feed (crude protein: 50.00%, crude fat: ≥4.00%, crude fiber: 7.00%, and phosphorus: 0.80%) three times per day to satiation. At 45 and 90 DPH, 10 fingerlings were randomly selected from each treatment and their total length (TL) and body weight (BW) were recorded to determine growth performance. Further, survival was calculated at each sampling period.

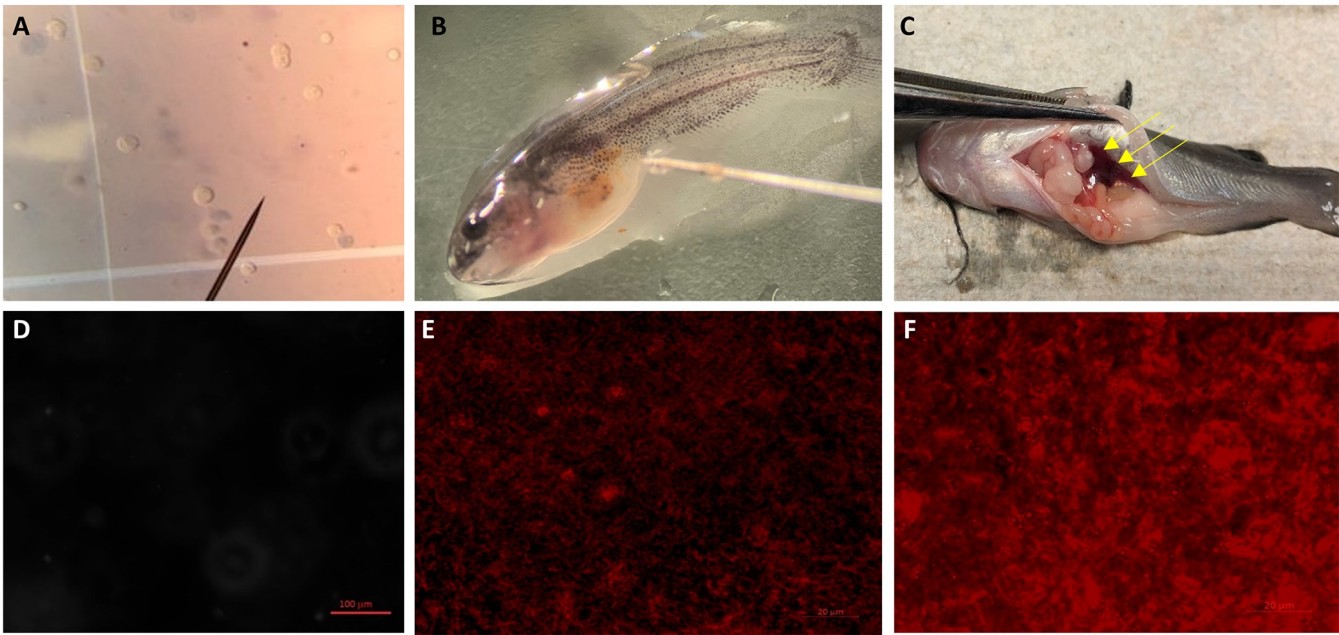

**Fig 1.** Stem cell (A) transplantation (intraperitoneally) (B) of donor derived stem cells into triploid white catfish (*Ameiurus catus*). Transplanted stem cells migrate to the genital ridge of the recipient, are incorporated, and initiate oogenesis or spermatogenesis. Gonadal growth of the xenogeny (C). The non-injected control treatment (D) showed no fluorescence, while those injected with channel catfish (*Ictalurus punctatus*) stem cells were fluorescing in fry sampled at 45 (E) and 90 DPH (F).

## Sample analyses at 45 and 90 DPH

Nine fingerlings were randomly selected from each treatment to evaluate the proliferation and colonization of transplanted BSCs and CSCs inside the triploid white catfish. The fingerlings were sacrificed via pithing and their gonads were surgically extracted. Each gonadal section was placed on a sterile microscope slide (1600221, Life Technologies) and covered with a coverslip. After slide preparation, digital images were taken using a Zeiss Imager A2 microscope equipped with a camera (Axio-cam 202) and Zen Pro v.6.1 software (Zeiss, Oberkochen, Germany). All samples containing donor-derived 'positive' cells exhibited a red fluorescence, whereas 'negative' cells did not display any fluorescence. Six fluorescent images from each gonad were analyzed using ImageJ software and cell number, cell area, cluster number, and cluster area were determined. Cells were categorized as having a fluorescence area of $<150\ \mu m^2$, while clusters were categorized by a fluorescence area $>150\ \mu m^2$.

## DNA extraction and PCR analysis

At each sampling period (45 DPH and 90 DPH), 5 fingerlings were randomly selected for DNA extraction and PCR analysis. Isolated gonad samples were transferred to 1.5 mL Eppendorf tubes and subsequently stored at -80°C until DNA extraction. DNA was extracted using proteinase K digestion followed by protein and ethanol precipitation [30] (Table 1).

**Table 1. Primers used for genes [Fst (follistatin) and Hamp (hepcidin antimicrobial protein)] to differentiate channel catfish (*Ictalurus punctatus*) and blue catfish (*I. furcatus*).**

| Gene | Forward primer | Reverse primer | Amplicon (bp) | |
|------|---------------|----------------|---------|------|
| | | | Channel | Blue |
| *Fst* | ATAGATGTAGAGGAGCATTTGAG | GTAACACTGCTGTACGGTTGAG | 348 | 399 |
| *Hamp* | ATACACCGAGGTGGAAAAGG | AAACAGAAATGGAGGCTGGAC | 222 | 262 |

### Statistical analyses

Breakpoint analysis was conducted with the Segmented package in R [31, 32]. The segmented regression or breakpoint analysis was conducted to identify the most suitable age for transplanting BSCs and CSCs into triploid white catfish hosts, taking into consideration factors such as offspring growth, survival, and fluorescent image evaluations.

## Results

### Growth and survival

No significant differences in TL and BW were obtained for transplanted BSCs and CSCs and control treatments at 45 and 90 DPH (all P > 0.097) (Fig 2). Segmented linear regression

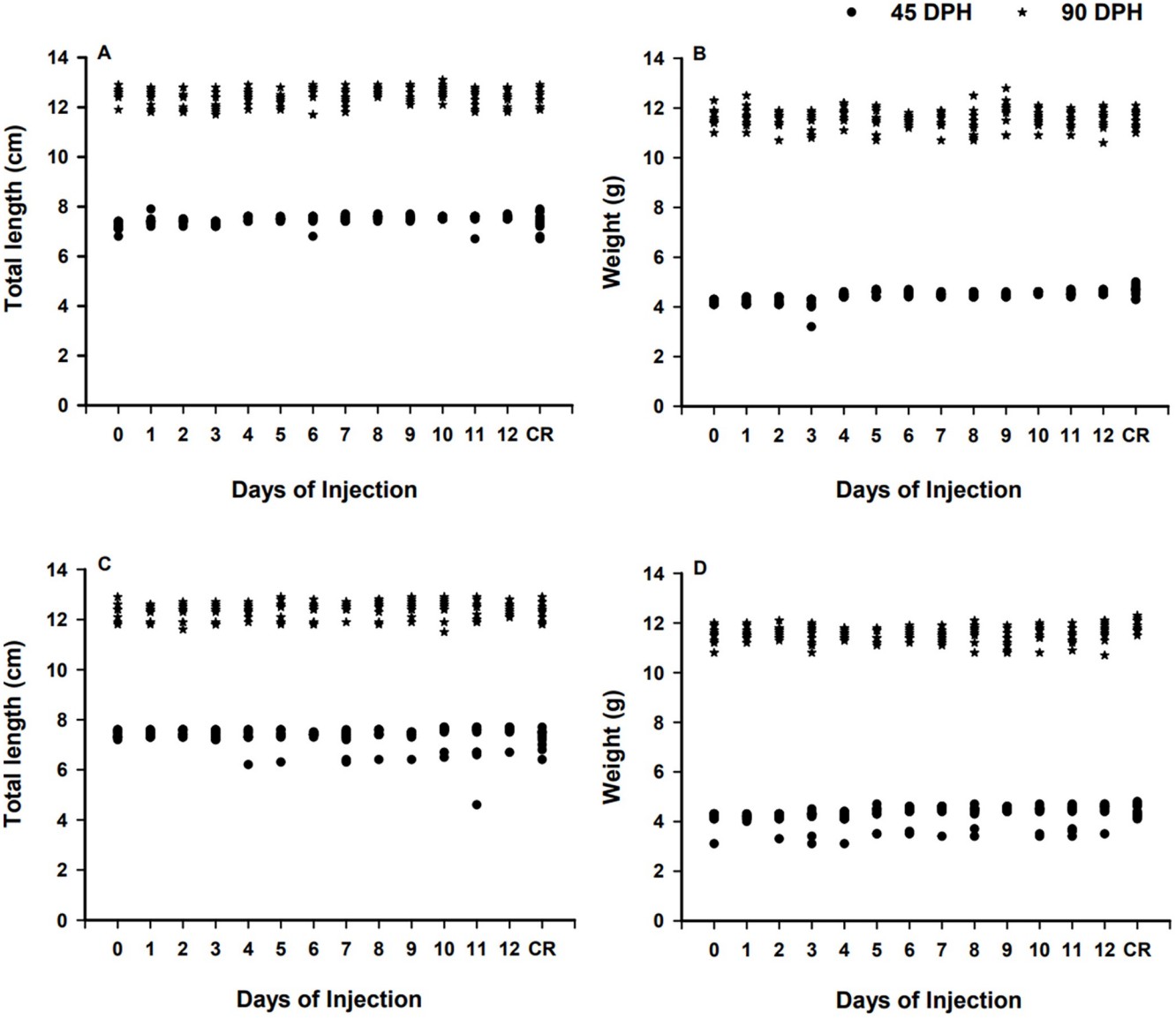

**Fig 2.** Total length (A, C) and weight (B, D) at 45 (●) and 90 (★) days post-hatch (DPH) of triploid xenogenic white catfish (*Ameiurus catus*) injected with blue catfish (*Ictalurus furcatus;* A, B) or channel catfish (*I. punctatus;* C, D) stem cells from 0 to 12 DPH. "CR" represents the non-injected control. Days of injection = Days post hatch.

showed that survival at 45 DPH of recipient fish injected with BSCs was increasing from 0 to 4.0 DPH ($P < 0.0001$, $R^2 = 0.72$, $y = 42.7 + 7.0x$; Fig 3A) and the same trend was observed for those injected with CSCs from 0 to 4.5 DPH ($P < 0.0001$, $R^2 = 0.86$, $y = 33.2 + 11.0x$; Fig 3C). After 4.0 DPH and 4.5 DPH, survival remained constant ($\geq 87.3\% \pm 2.9$; $P = 0.134$) in BSC and CSC transplanted treatments, respectively. A similar fluctuation was observed at 90 DPH, as survival gradually increased in the recipient fish when injected with BSCs from 0 to 5.5 DPH ($P < 0.0001$, $R^2 = 0.93$, $y = 19.76 + 9.41x$; Fig 3B) and with CSCs from 0 to 4.5 DPH ($P < 0.0001$, $R^2 = 0.91$, $y = 22.7 + 9.56x$; Fig 3D). Thereafter, survival remained high ($\geq 81.2\% \pm 2.1$; $P = 0.154$) for BSCs and CSCs fry injected until 12 DPH.

## Quantifying fluorescent labeling

Day of stem cell injection (0 to 12 DPH) had a significant impact on stem cell proliferation and colonization in the surrogate fish. Briefly, at 45 DPH, cell area and cluster area were

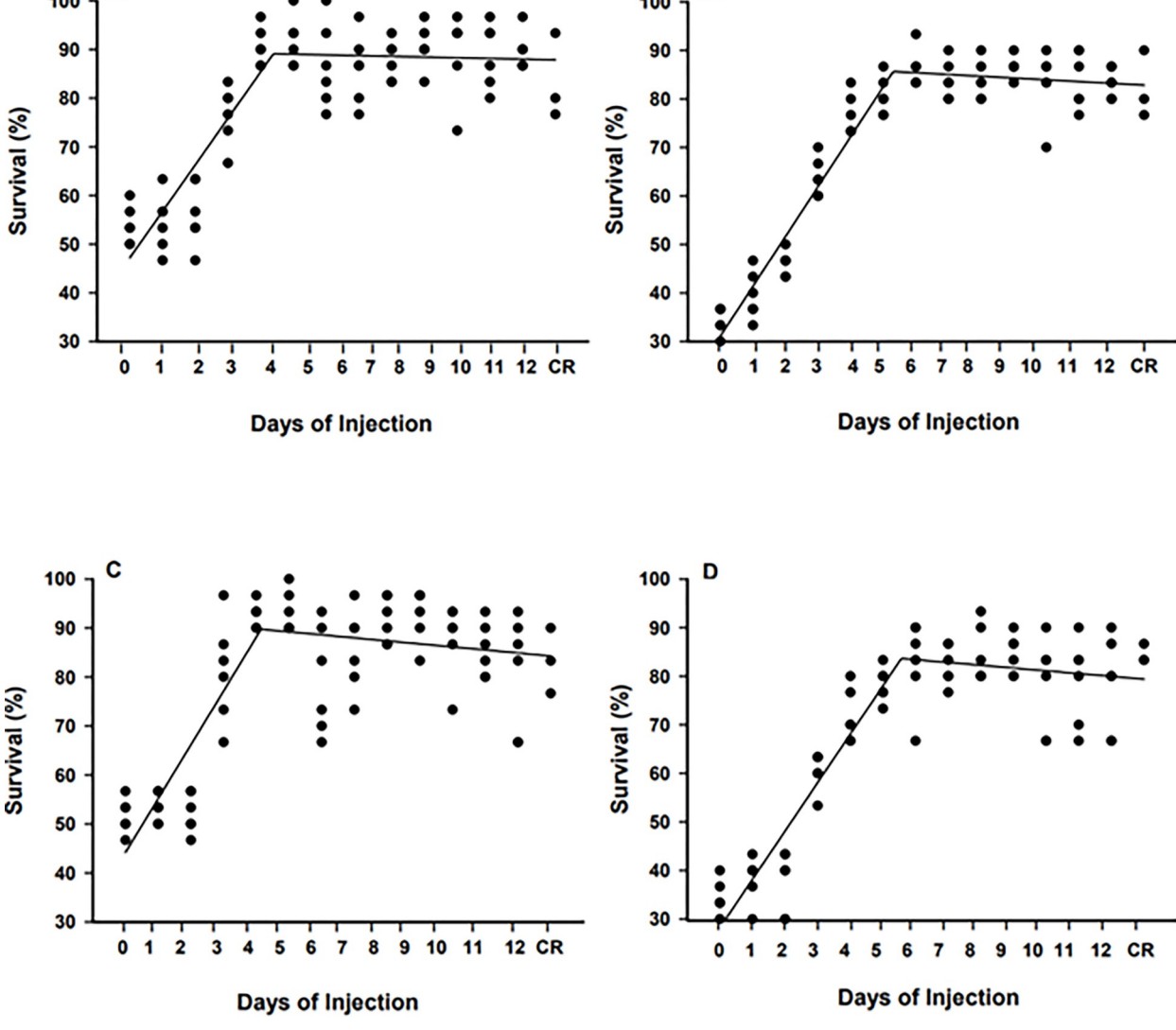

**Fig 3.** Percent survival at 45 (A, C) and 90 days post-hatch DPH (B, D) of triploid xenogenic white catfish (*Ameiurus catus*) fry injected with blue catfish (*Ictalurus furcatus*; A, B) or channel catfish (*I. punctatus*; C, D) stem cells from 0 to 12 days post-hatch (DPH). "CR" represents the non-injected control. Days of injection = Days post hatch.

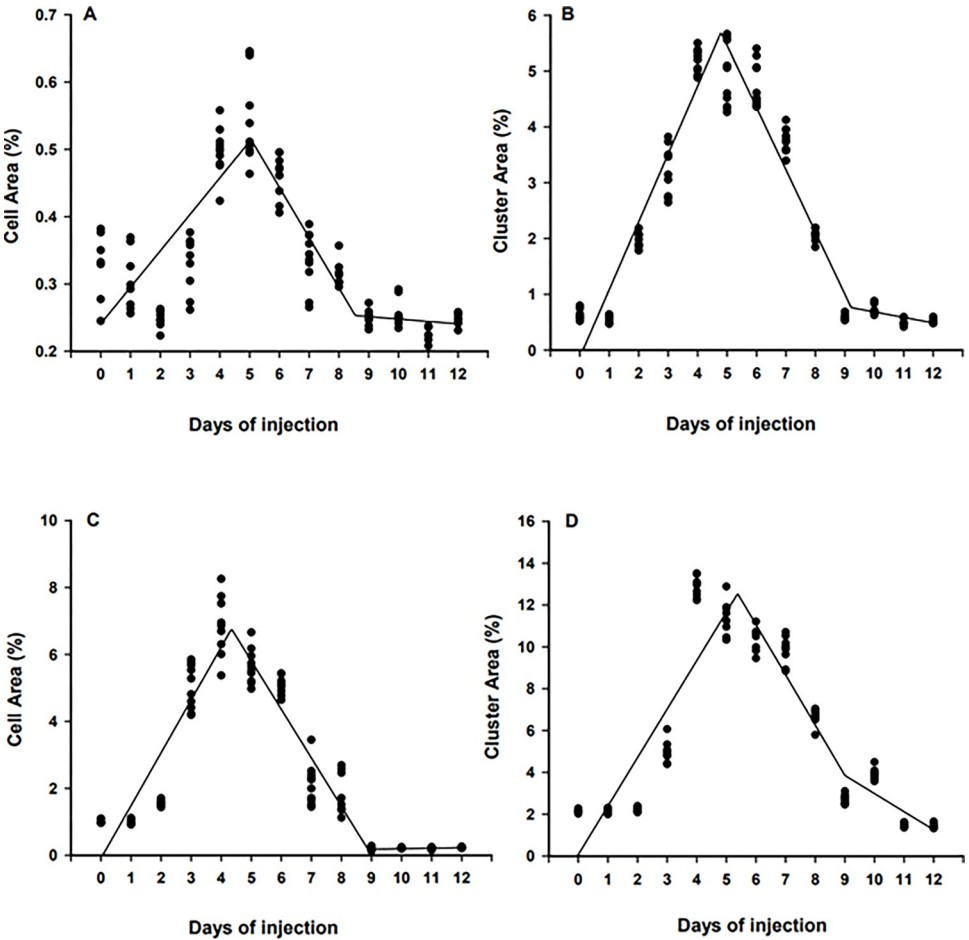

**Fig 4.** Percent cell area and cluster area at 45 (A, B) and 90 days post-hatch (DPH; C, D) of triploid white catfish (*Ameiurus catus*) fry injected with blue catfish (*Ictalurus furcatus*) stem cells from 0 to 12 DPH. Days of injection = Days post hatch.

increasing for individuals injected from 0 to 5 DPH ($P < 0.0001$, $R^2 = 0.58$, y = 0.18 + 0.054x; Fig 4A) and from 0 to 4.8 DPH ($P < 0.0001$, $R^2 = 0.89$, y = -0.94 + 1.05x; Fig 4B), respectively in transplanted fish receiving BSCs. After reaching the peak levels, both cell area ($P < 0.0001$, $R^2 = 0.89$, y = 6.56–1.11x; Fig 4A) and cluster area ($P < 0.0001$, $R^2 = 0.88$, y = 6.57–1.12x; Fig 4B) gradually declined with no further decrease after 8.6 DPH and 9.2 DPH days of injection, respectively. A similar trend was detected at 90 DPH. Briefly, the percentage cell area and cluster area increased daily for fish injected from 0 to 4.2 DPH ($P < 0.0001$, $R^2 = 0.81$, y = -1.67 + 1.58x; Fig 4C) and from 0 to 5.2 DPH ($P < 0.0001$, $R^2 = 0.76$, y = -2.20 + 2.30x; Fig 4D), respectively. After reaching peak levels, both cell area ($P < 0.0001$, $R^2 = 0.91$, y = 8.34–1.37x; Fig 4C) and cluster area ($P < 0.0001$, $R^2 = 0.84$, y = 14.47–2.11x; Fig 4D) gradually decreased.

Similarly, the day of stem cell injection (0 to 12 DPH) had a significant impact on CSCs proliferation and colonization in triploid white catfish. For instance, at the 45 DPH sampling period, cell area and cluster area in recipient fish was increasing for individuals injected from 0 to 4.9 DPH ($P < 0.0001$, $R^2 = 0.59$, y = 0.11 + 0.04x; Fig 5A) and from 0 to 4.8 DPH ($P < 0.0001$, $R^2 = 0.87$, y = 0.03 + 0.78x; Fig 5B), respectively. After reaching peak levels, the percentage of fluorescent cell area ($P < 0.0001$, $R^2 = 0.82$, y = 0.39–0.05x; Fig 5A) and cluster area ($P < 0.0001$, $R^2 = 0.91$, y = 5.65–0.81x; Fig 5B) in the surrogate declined with no further

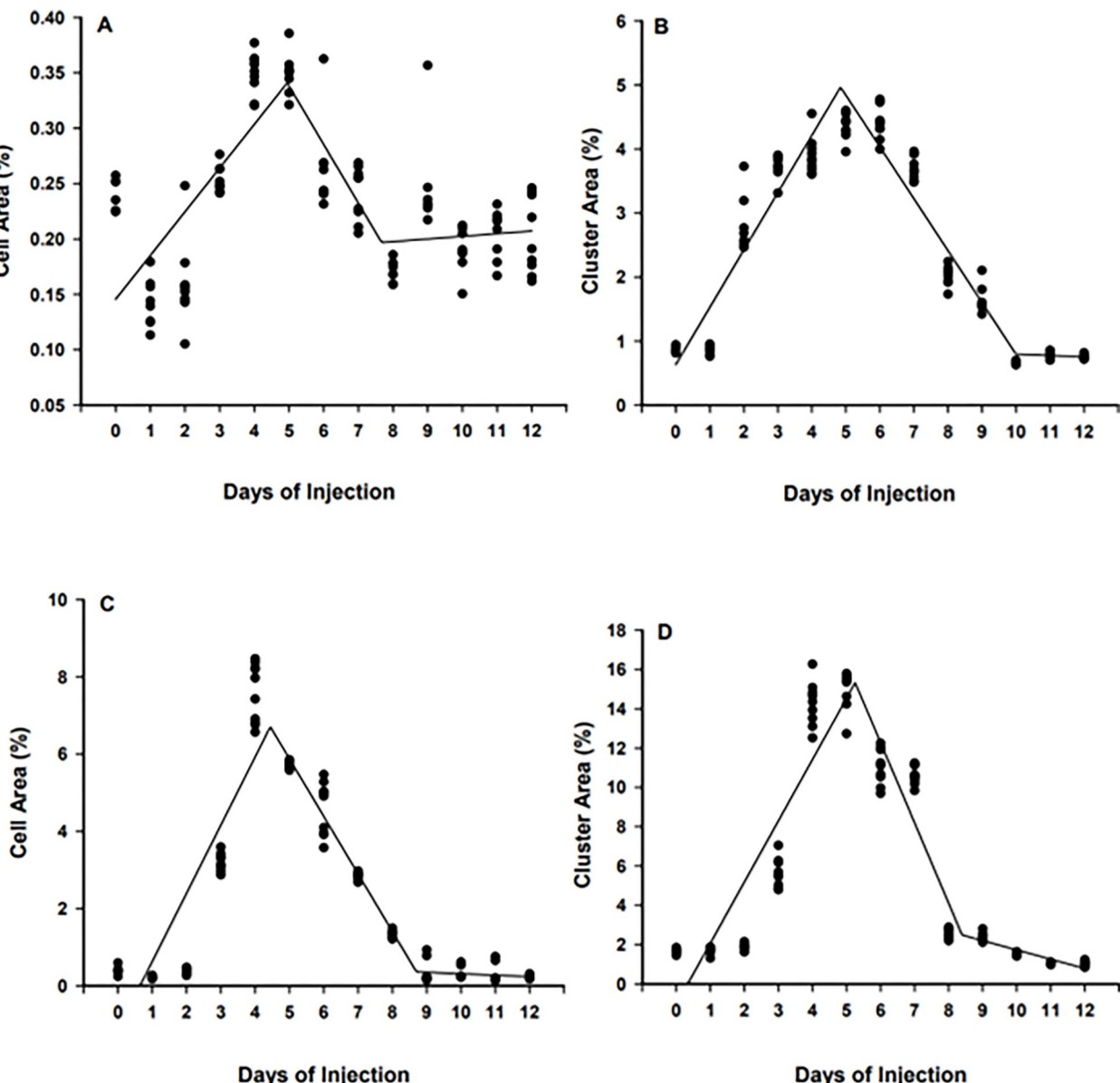

**Fig 5.** Percent cell area and cluster area at 45 (A, B) and 90 days post-hatch (DPH; C, D) of triploid white catfish (*Ameiurus catus*) fry injected with channel catfish (*Ictalurus punctatus*) stem cells from 0 to 12 DPH. Days of Injection = Days post hatch.

decrease after 7.6 DPH and 10.0 DPH, respectively. Similarly, at 90 DPH, the percentage cell area and cluster area gradually increased from 0 to 4.4 DPH ($P < 0.0001$, $R^2 = 0.76$, y = - 2.89 + 1.76x; Fig 5C) and from 0 to 5.1 DPH ($P < 0.0001$, $R^2 = 0.82$, y = - 4.14 + 3.11x; Fig 5D), respectively. Thereafter, cell area ($P < 0.0001$, $R^2 = 0.95$, y = 9.12–1.53x; Fig 5C) and cluster area ($P < 0.0001$, $R^2 = 0.83$, y = 19.08–3.66x; Fig 5D) declined in the host with no further decrease after 8.8 DPH and 8.3 DPH, respectively.

## PCR analysis

PCR analysis indicated that recipient white catfish fry possessed DNA from transplanted blue catfish donor-derived stem cells in their gonads at 45 and 90 DPH (Fig 6). At 45 DPH, both

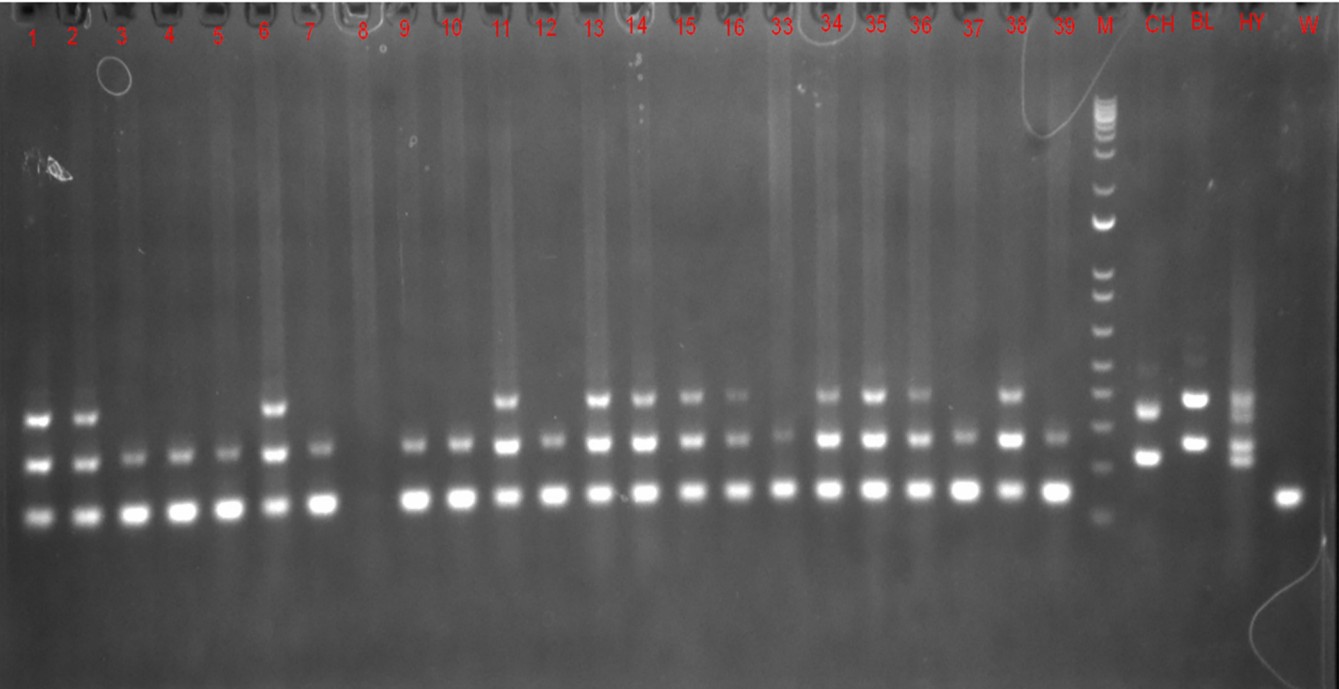

**Fig 6. Sample results from PCR for detecting blue catfish (*Ictalurus furcatus*) donor cells in the testes of triploid white catfish (*I. punctatus*).** Blue catfish and white catfish cells were differentiated with PCR using follistatin (Fst) and hepcidin antimicrobial protein (Hamp) genes as markers. W = white catfish control, BL = blue catfish control, CH = channel catfish control, HY = channel -blue hybrid, M = Marker.

BSCs and CSCs had the highest percentage of xenogens when recipient fish were injected from 4 to 5 DPH (all 80.0%) and 3.0 to 5.0 DPH (80.0%), respectively. At the 90 DPH, the highest percentage of xenogens was detected from 4.0 to 6.0 DPH in both BSCs and CSCs transplanted treatments.

## Discussion

The potential of hybrid catfish embryo production through xenogenesis has been highlighted in the literature by transplanting BSCs into a triploid channel catfish host [12, 15–17]. However, having a surrogate with a short maturation time and a smaller body size than channel catfish would be ideal for commercial applications. Hence, filling this research gap, our current study identified the suitability of white catfish as a surrogate for hybrid catfish production and identified 4.0 to 6.0 DPH as the most suitable age to transplant donor-derived stem cells into a sterile white catfish recipient, based on growth, survival, cell proliferation, and colonization.

The fluorescent cell linker dye PKH26 offers the longest *in vivo* half-life, around 100 days, which makes it ideal for in *vivo* cell tracking such as in cell proliferation, and colonization studies [33, 34]. Thus, it has been successfully utilized in staining germ cells in several cell transplantation studies [35–39]. In the present study, we observed fluorescently labeled donor-derived BSCs and CSCs in white catfish (surrogate) gonads at 45 and 90 DPH after intraperitoneal transplantation, which indicates the capability of colonization and proliferation of donor-derived gametes. Further, both cell and cluster areas significantly increased from 45 to 90 DPH, clearly indicating proliferation and colonization of the donor-derived cells. Specifically, segmented linear regression demonstrated higher cell and cluster colonization when donor-derived stem cells were injected into recipient fish between 4.0 to 6.0 DPH. A similar trend

was detected in our previous study which was conducted using channel catfish as the surrogate where significantly higher cell and cluster colonization was noted when injected into recipient fish between 4.0 to 6.0 DPH [16]. To validate the success of hybrid catfish embryo production through xenogenesis, xenogenic white catfish have been utilized (unpublished data). In 2019, a xenogenic white catfish pair that had been transplanted with 80,000 unsorted channel catfish stem cells at 3DPH, produced channel catfish progeny. Similarly, in both 2020 and 2021, pairs of xenogenic white catfish that were transplanted with unsorted channel catfish stem cells at 3 and 4 DPH successfully generated channel catfish offspring. Additionally in 2022, three blue catfish progenies were produced by xenogenic white catfish pairs that received 40,000 unsorted blue catfish stem cells between 4 and 5 DPH.

According to Dunham et al. [39], white catfish grow more rapidly as fingerlings than channel catfish. The growth performance differences at their fingerling stage could be the reason for slight time differences among white and channel catfish. According to the resulting data, the average TL and BW of channel catfish at 50 DPH was 5.1 cm and 2.46 g, respectively. For the white catfish, it was 7.5 cm and 4.45 g, respectively. The faster growth rate of white catfish at the fingerling stage could be an added advantage during the stem cell transplantation procedure since there are some practical challenges to injecting fry during this early small life stage, as it is difficult to insert the needle into the small body cavity between the anal opening and yolk-sac (i.e., where genital ridge is expected to form). Thus, having a higher growth rate can mitigate this practical issue, increasing the suitability of white catfish as a surrogate species.

Yoshizaki et al. [40] indicated that a potential obstacle to donor cell proliferation and colonization could be the rejection of transplanted cells by the immune system of the recipient fish. To overcome this potential obstacle, newly hatched fry were used as recipients in several studies due to their immature immune system immediately after hatching [41–47]. This could be a potential reason for the cluster and cell area which gradually decreased after 5 DPH with the fish growth. However, lower cluster and cell areas were detected between 0–3 DPH, which could be due to technical challenges to transplanting/injecting cells to fry during this early small life stage or maybe it is not a good development stage for colonization and proliferation of transplanted stem cells. Transplanting stem cells between 4 to 6 DPH would be an added advantage. By this time, white catfish fry have absorbed their yolk-sac facilitating easy needle insertion by targeting the genital ridge. Moreover, fry at 4 to 6 DPH are more tolerant of physical damage compared to newly hatched fry, during the cell transplantation through the needle. In the present study, there was no significant difference in survival rates between the best treatments (4.0 and 6.0 DPH) and the control treatment ($\geq$81.2% ± 2.1; P = 0.134). A comparable survival rate ($\geq$82%) was also identified in our previous study [16], where channel catfish was used as the surrogate in recirculating aquaculture systems (RAS).

Even though there are several desirable characteristics for white catfish as a surrogate species, still there are some drawbacks. Low survival (personal observation) and low seinability [48] of white catfish during the pond culture, are the major challenges to overcome for commercial application. While xenogenic white catfish exhibited a higher survival rate during the current study, consistent with previous studies for channel catfish [16], their survival declines upon reaching sexual maturity due to aggressive behaviors. Especially in pond culture, white catfish have a low survival rate and low seinability/harvestability. Additionally, like many other *ictalurid* catfish, white catfish are not totally resistant to several bacterial, parasite, and viral diseases such as columnaris (causative agent; *Flexibacter columnare*), enteric septicemia of catfish (*Edwardsiella ictaluri*), aeromonas disease (*Aeromonas hydrophila*), ich (*Ichthyophthirius multifilis*), and channel catfish virus. Further, white catfish males are aggressive and fight with each other when held communally [13]. This results in frequent wounds in males, which leads to secondary infections and increased disease susceptibility.

Even though seinability is considered an undervalued trait in ponds [23], in the case of white catfish, more difficult seinability could be a major inconvenience, causing extra seining effort. Culturing white catfish in RAS could be an alternative option to mitigate these harvestability and disease resistance issues. In the case of RAS, fighting can be controlled or reduced by providing shelters (this has not been studied in ponds, but sheltering should be possible in ponds). Additionally, excessively aggressive, or passive males could be isolated.

In conclusion, the current study demonstrated the suitability of white catfish as a surrogate species by transplanting stem cells at 4.0 to 6.0 DPH, and these findings can be used to enhance the efficiency of germ cell transplantation in xenogenesis for commercial-scale hybrid catfish production.

## Supporting information

**S1 Raw image.**
(TIF)

## Acknowledgments

We would like to acknowledge the Catfish Genetics Research Unit staff and the EW Shell Fisheries Center at Auburn University for fish care, labor, and maintenance.

## Author Contributions

**Conceptualization:** Darshika Udari Hettiarachchi.

**Data curation:** Darshika Udari Hettiarachchi, Ian A. E. Butts.

**Formal analysis:** Darshika Udari Hettiarachchi, Veronica N. Alston, Logan Bern, Jacob Al-Armanazi, Baofeng Su, Mei Shang, Jinhai Wang, Rex A. Dunham.

**Funding acquisition:** Ian A. E. Butts.

**Investigation:** Rex A. Dunham, Ian A. E. Butts.

**Methodology:** Darshika Udari Hettiarachchi, De Xing, Shangjia Li, Rex A. Dunham, Ian A. E. Butts.

**Project administration:** Rex A. Dunham, Ian A. E. Butts.

**Resources:** Darshika Udari Hettiarachchi, Rex A. Dunham, Ian A. E. Butts.

**Software:** Matthew K. Litvak, Ian A. E. Butts.

**Supervision:** Matthew K. Litvak, Rex A. Dunham, Ian A. E. Butts.

**Visualization:** Ian A. E. Butts.

**Writing – original draft:** Darshika Udari Hettiarachchi.

**Writing – review & editing:** Darshika Udari Hettiarachchi, Rex A. Dunham, Ian A. E. Butts.

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
