## [Decision Letter · Decision Letter 0]

14 Feb 2024

PONE-D-23-35581Production of xenogenic catfish by transplanting blue catfish (Ictalurus furcatus) and channel catfish (I. punctatus) stem cells into white catfish (Ameiurus catus) triploid fryPLOS ONE

Dear Dr. Hettiarachchi,

Thank you for submitting your manuscript to PLOS ONE. After careful consideration, we feel that it has merit but does not fully meet PLOS ONE’s publication criteria as it currently stands. Therefore, we invite you to submit a revised version of the manuscript that addresses the points raised during the review process.

We look forward to receiving your revised manuscript.

Kind regards,

A. K. Shakur Ahammad, PhD

Academic Editor

PLOS ONE

 [Agriculture and Food Research Initiative Competitive Grant no. 2018-67015-27614 from the USDA National Institute of Food and Agriculture].  

[This project was supported by Agriculture and Food Research Initiative Competitive Grant no. 2018-67015-27614 from the USDA National Institute of Food and Agriculture. We would like to acknowledge the Catfish Genetics Research Unit staff and the EW Shell Fisheries Center at Auburn University for fish care, labor, and maintenance.]

 [Agriculture and Food Research Initiative Competitive Grant no. 2018-67015-27614 from the USDA National Institute of Food and Agriculture]

5. Please provide a complete Data Availability Statement in the submission form, ensuring you include all necessary access information or a reason for why you are unable to make your data freely accessible. If your research concerns only data provided within your submission, please write "All data are in the manuscript and/or supporting information files" as your Data Availability Statement.

5. Please amend the manuscript submission data (via Edit Submission) to include author De Xinga and Shangjia Li.

7. We note that Figure 1 and 2 in your submission contain copyrighted images. All PLOS content is published under the Creative Commons Attribution License (CC BY 4.0), which means that the manuscript, images, and Supporting Information files will be freely available online, and any third party is permitted to access, download, copy, distribute, and use these materials in any way, even commercially, with proper attribution. For more information, see our copyright guidelines: http://journals.plos.org/plosone/s/licenses-and-copyright.

1. You may seek permission from the original copyright holder of Figure 1 and 2 to publish the content specifically under the CC BY 4.0 license. 

8. Please include a copy of Table 1 which you refer to in your text on page 21.

Additional Editor Comments:

1) Figure Labeling Improvement: The authors should consider refining figure labels for clarity, ensuring that descriptions in the Results section align accurately with the presented figures. This will enhance the reader's understanding of the data.

2) Discussion Enhancement - Host Fish Selection Criteria: To provide a more comprehensive discussion, the authors could elucidate the criteria that guided their choice of white catfish as the host species for xenogenesis. Clarifying the rationale behind selecting this particular host can offer valuable insights and context to readers.

3) Comparison with Previous Data: Strengthen the Discussion by comparing the current study's survival rates and xenogenesis efficiency with any available data on blue and channel catfish. This comparative analysis can highlight advancements or variations in the present research, contributing to a more robust understanding of the findings.

4) Explanation of Transplantation Efficiency Peaks: Elaborate on the reasons behind the observed peak in transplantation efficiency between 4 and 5 days post-hatching (DPH). Providing a detailed explanation for this timeframe could offer valuable insights into the biological mechanisms or environmental factors influencing the success of xenogenesis during this specific developmental stage.

By addressing these aspects, the manuscript can be refined to enhance clarity, context, and the overall contribution of the research.

Reviewers' comments:

Reviewer's Responses to Questions

**Comments to the Author**

1. Is the manuscript technically sound, and do the data support the conclusions?

Reviewer #1: Yes

Reviewer #2: Yes

2. Has the statistical analysis been performed appropriately and rigorously? 

Reviewer #1: Yes

Reviewer #2: Yes

3. Have the authors made all data underlying the findings in their manuscript fully available?

Reviewer #1: Yes

Reviewer #2: Yes

4. Is the manuscript presented in an intelligible fashion and written in standard English?

Reviewer #1: Yes

Reviewer #2: Yes

5. Review Comments to the Author

Reviewer #1: Reviewer comments on the Manuscript

As a reviewer my comments on the Manuscript narrated below for further improvement of the manuscript narrated below-

1.Advise about the title: Authors can think about the title alternatively appropriate like ”Advancing Aquaculture: Xenogenic Catfish Production through Blue and Channel Catfish Stem Cell Transplants in White Catfish (Ameiurus catus) Triploid Fry”

2.The abstract clearly outlines the objective of the study, which is to assess the effectiveness of triploid white catfish as a host for the transplantation of blue and channel catfish stem cells. It might be beneficial to include brief insights into the potential implications of these findings for commercial aquaculture and any challenges or limitations encountered during the study.

3.In introduction mentions that xenogenesis can mitigate challenges in hybrid catfish production, but it would be helpful to elaborate on why xenogenesis is considered a promising alternative. Highlight the specific advantages and potential improvements it offers over existing methods.

Check the consistency in terminology such as "xenogenesis" and "xenogenic." Make sure these terms are used uniformly throughout the introduction for clarity.

Clarify the rationale behind choosing triploid white catfish as the host species for xenogenesis. Explain how the unique characteristics of white catfish make it advantageous for the xenogenic process compared to channel catfish.

Addressing these points can help strengthen the introduction and provide a clearer context for the study, making it more accessible to a wider audience

4.The methodology is quite detailed, which is good for transparency. However, breaking it down into subsections with clear headings could enhance readability. For instance, separate sections for broodstock management, stem cell extraction, labeling, transplantation, and sample analysis would make it easier for readers to follow.

Details on the environmental conditions of the aquaculture system, such as water temperature, pH levels, and any other relevant parameters, are important for understanding the context of the study. Including information on these conditions can help readers assess the validity and generalizability of the findings.

It's mentioned that breakpoint analysis was conducted, but more details on the specific statistical tests or models used, as well as the criteria for determining breakpoints, would enhance the rigor of the methodology.

5.In the result section, the results are presented in a clear and organized manner. The use of figures (e.g., Fig. 3, Fig. 4) enhances the visual representation of key findings. However, it may be helpful to include concise captions for each figure to guide readers through the data.

The description of how the day of stem cell injection impacts stem cell behavior over time is well articulated. The temporal analysis provides valuable insights into the dynamics of cell and cluster areas. Ensure that the text explains the biological significance of the observed patterns, especially the peak levels and subsequent declines.

Note the consistency in the trends observed between the two types of stem cells (blue catfish and channel catfish) and how these trends compare at different time points. This helps build confidence in the reliability and reproducibility of the experimental outcomes.

Continue to interpret the data in the context of the study's objectives and the broader implications for xenogenic catfish production. Discuss how the observed trends align with the goals of the experiment and any potential applications or challenges identified.

Consider including a brief section or concluding remarks discussing potential implications of the results for future research or practical applications in aquaculture.

6.The discussion is clear and well-organized, addressing the key findings step by step. However, consider providing a brief overview or roadmap at the beginning of the discussion to guide readers through the main points and structure of the discussion.

How was the specific age range of 4.0 to 5.0 days post-hatch (DPH) identified as optimal for stem cell transplantation into white catfish hosts, and what factors were considered in making this determination?

The discussion acknowledges technical challenges during the early stages of stem cell transplantation. Can you elaborate on specific challenges faced during injection into fry and whether these challenges are inherent to the procedure or related to the life stage of the fish?

The discussion suggests recirculating aquaculture systems (RAS) as a potential alternative. Could you delve deeper into how RAS could overcome the identified issues of harvestability and disease resistance in white catfish, and are there any specific challenges associated with this alternative?

7.Is there any ethical consideration permission required for this study ? or not applicable in this case?

Reviewer #2: Title: Production of xenogenic catfish by transplanting blue catfish (Ictalurus furcatus) and channel catfish (I. punctatus) stem cells into white catfish (Ameiurus catus) triploid fry

General Comments to the Authors:

The paper assessed the effectiveness of white catfish as host species to transplant blue catfish and channel catfish stem cells through xenogenesis. The authors can improve the manuscript by improving the labels in the figures, making sure that the values described in the Results section coincide with those in the figures. The Discussion can be improved by including pertinent information such as the basis/criteria for choosing the host fish for xenogenesis, compare the present data on survival of xenogens and efficiency of xenogenesis with previous data with blue and channel catfish (if any), and explain the reasons why the transplantation efficiency was greatest between 4 and 5 DPH.

Please see attached file for detailed review.

6. PLOS authors have the option to publish the peer review history of their article (what does this mean?). If published, this will include your full peer review and any attached files.

Reviewer #1: No

Reviewer #2: No

---

## [Editor Report · Decision Letter 1]

8 Apr 2024

Advancing Aquaculture: Production of xenogenic catfish by transplanting blue catfish (Ictalurus furcatus) and channel catfish (I. punctatus) stem cells into white catfish (Ameiurus catus) triploid fry

PONE-D-23-35581R1

Dear Dr. Hettiarachchi,

We’re pleased to inform you that your manuscript has been judged scientifically suitable for publication and will be formally accepted for publication once it meets all outstanding technical requirements.

Kind regards,

A. K. Shakur Ahammad, PhD

Academic Editor

PLOS ONE

Additional Editor Comments (optional):

Congratulations to authors.
---

## [Editor Report · Acceptance letter]

1 May 2024

PONE-D-23-35581R1 

PLOS ONE

Dear Dr. Hettiarachchi, 

I'm pleased to inform you that your manuscript has been deemed suitable for publication in PLOS ONE. Congratulations! Your manuscript is now being handed over to our production team.

Kind regards, 

on behalf of

Dr. A. K. Shakur Ahammad 

Academic Editor

PLOS ONE